


**Reviews and Syntheses: Changing ecosystem influences on soil thermal regimes in northern**
**high-latitude permafrost regions**
Michael M. Loranty[1], Benjamin W. Abbott[2], Daan Blok[3], Thomas A. Douglas[4], Howard E.
Epstein[5], Bruce C. Forbes[6], Benjamin M. Jones[7], Alexander L. Kholodov[8], Heather Kropp[1], Avni
Malhotra[9], Steven D. Mamet[10], Isla H. Myers-Smith[11], Susan M. Natali[12], Jonathan A
O'Donnell[13], Gareth K. Phoenix[14], Adrian V. Rocha[15], Oliver Sonnentag[16], Ken D. Tape[17],
Donald A. Walker[18]
[1]Department of Geography, Colgate University, Hamilton, NY 13346 USA
[2]Department of Plant and Wildlife Sciences, Brigham Young University, Provo, UT 84602 USA
[3]Department of Physical Geography and Ecosystem Science, Lund University, S-223 62 Lund,
Sweden
[4]U.S. Army Cold Regions Research and Engineering Laboratory Fort Wainwright, Alaska 99703
USA
[5]Department of Environmental Sciences, University of Virginia, Charlottesville, VA 22904 USA
[6]Arctic Centre, University of Lapland, FI-96101, Rovaniemi, Finland
[7] U.S. Geological Survey Alaska Science Center, Anchorage, AK 99508 USA
[8]Geophysical Institute, University of Alaska Fairbanks, Fairbanks, AK 99775 USA
[9]Environmental Sciences Division and Climate Change Science Institute, Oak Ridge National
Laboratory, Oak Ridge, TN 37831-6301, USA
[10]Department of Soil Science, University of Saskatchewan, Saskatoon, SK S7N 5A8 Canada
[11]School of GeoSciences, University of Edinburgh, Edinburgh, UK
[12]Woods Hole Research Center, Falmouth, MA 02540 USA
[13]Arctic Network, National Park Service, Anchorage, AK 99501 USA
[14]Department of Animal and Plant Sciences, University of Sheffield, Western Bank, Sheffield,
S10 2TN, United Kingdom
[15]Department of Biological Sciences and the Environmental Change Initiative, University of
Notre Dame, Notre Dame 46556 USA
[16]Département de géographie, Université de Montréal, Montréal, QC H2V 3W8, Canada
[17]Institute of Northern Engineering, Water & Environmental Research Center, University of
Alaska, Fairbanks, AK 99775 USA
[18]Institute of Arctic Biology, University of Alaska Fairbanks, AK 99775 USA
Correspondence to: Michael M. Loranty, email  - mloranty@colgate.edu, phone – 315-228-6057



## Abstract

Permafrost soils in arctic and boreal ecosystems store twice the amount of current atmospheric carbon that may be mobilized and released to the atmosphere as greenhouse gases when soils thaw under a warming climate. This permafrost carbon climate feedback is among the most globally important terrestrial biosphere feedbacks to climate warming, yet its magnitude remains highly uncertain. This uncertainty lies in predicting the rates and spatial extent of permafrost thaw and subsequent carbon cycle processes. Terrestrial ecosystem influences on surface energy partitioning exert strong control on permafrost soil thermal dynamics and are critical for understanding permafrost soil responses to climate change and disturbance. Here we review how arctic and boreal ecosystem processes influence permafrost soils and characterize key ecosystem changes that regulate permafrost responses to climate. While many of the ecosystem characteristics and processes affecting soil thermal dynamics have been examined in isolation, interactions between processes are less well understood. In particular connections between vegetation, soil moisture, and soil thermal properties affecting permafrost conditions could benefit from additional research. In particular, connections between vegetation, soil moisture, and soil thermal properties affecting permafrost could benefit from additional research. Changes in ecosystem distribution and vegetation characteristics will alter spatial patterns of interactions between climate and permafrost. In addition to shrub expansion, other vegetation responses to changes in climate and disturbance regimes will all affect ecosystem surface energy partitioning in ways that are important for permafrost. Lastly, changes in vegetation and ecosystem distribution will lead to regional and global biophysical and biogeochemical climate feedbacks that may compound or offset local impacts on permafrost soils. Consequently, accurate prediction of the permafrost carbon climate feedback will require detailed





understanding of changes in terrestrial ecosystem distribution and function and the net effects of
multiple feedback processes operating across scales in space and time.



## 1 Introduction


Permafrost is perennially frozen ground that underlies approximately 24% of northern
hemisphere land masses, primarily in arctic and boreal regions (Brown *et al.*, 1998). Soils in
permafrost ecosystems have a seasonally thawed active layer that develops each summer. Soil
organic carbon and nutrients stored in the active layer are subject to mineralization, uptake by
plants and microbes, and lateral hydrological transport, as components of contemporary
biogeochemical cycles. Carbon and nutrients locked in perennially frozen ground are
considerably less active, sometimes remaining isolated from global cycles for millions of years.
However, changes in temperature, associated with recent climatic change are warming soils in
many high-latitude regions (Romanovsky *et al.*, 2010), introducing permafrost carbon and
nutrients to modern biogeochemical cycles (Schuur *et al.*, 2015). Some carbon and nutrients may
be released to the atmosphere by microbial activity in the form of carbon dioxide, methane, and
nitrous oxide, greenhouse gases that contribute to further warming (e.g. Koven *et al.*, 2011;
Abbott & Jones, 2015; Voigt *et al.*, 2017). While the magnitude of this permafrost-climate
feedback remains uncertain, it is considered one of the largest terrestrial feedbacks to climate
change, potentially enhancing human-induced emissions by 22-40% by the end of the century
(Schuur *et al.*, 2013; 2015).
A major source of uncertainty in estimating the timing and magnitude of the permafrost
climate feedback is the complexity of the soil thermal response of permafrost ecosystems to
atmospheric warming. Permafrost soil temperature and its response to climatic change are highly
variable across space and time (Jorgenson *et al.*, 2010), owing to multiple biophysical
interactions that modulate soil thermal regimes across arctic and boreal regions (Romanovsky *et*
*al.*, 2010). In general, permafrost temperature decreases and permafrost thickness and spatial



extent increase along a northward climatic gradient. In more northern locations, the areal
distribution of permafrost may be continuous (> 90% areal extent), whereas at lower latitudes
discontinuous, sporadic, and isolated permafrost (> 50-90%, 10-50%, and < 10% areal extent,
respectively) (Brown *et al.*, 1998) have large areas that are not perennially frozen. This general
latitudinal gradient is interrupted by considerable local variability in active layer and permafrost
thickness and temperature due to differences in local climate, vegetation, soil properties,
hydrology, topography, and snow characteristics. These factors can exert positive and negative
effects on permafrost thermal state, mediating a high degree of spatial and temporal variability in
the relationship between air and permafrost soil temperatures (Shur & Jorgenson, 2007;
Jorgenson *et al.*, 2010). Understanding how ecosystem characteristics influence local and
regional permafrost temperature is critical to interpreting variability in rates of recent permafrost
temperature increases (Romanovsky *et al.*, 2010), and to predict the magnitude and timing of the
permafrost climate feedback. However, links between permafrost and climate could
fundamentally change as arctic and boreal vegetation (e.g. Pearson *et al.*, 2013) and disturbance
regimes (e.g. Kasischke & Turetsky, 2006) shift in response to climate.

Here, we review how ecosystem structural and functional properties influence permafrost

soil thermal dynamics in arctic and boreal regions. We focus on how ecosystem responses to a
changing climate alter the thermal balance of permafrost soils (energy moving into and out of
permafrost soil) and how these thermal dynamics translate into seasonal and interannual
temperature shifts. Our objectives are to 1) identify and review the key mechanisms by which
terrestrial ecosystem structure and function influence permafrost soil thermal dynamics; 2)
characterize changes in these ecosystem properties associated with altered climate and
disturbance regimes; 3) identify and characterize potential feedbacks and uncertainties arising



from multiple opposing processes operating across spatial and temporal scales; and 4) identify
key challenges and research questions that need to be addressed to better constrain how
continued climate-mediated ecosystem changes will affect soil thermal dynamics in the
permafrost zone.

**2 Ecosystem influences on permafrost soil thermal dynamics**
Soil thermal dynamics in the permafrost zone are governed by ground-atmosphere energy
exchange and internal energy transfers. The simplified thermal balance at the ground surface is
the difference between net radiation ($R_N$) absorbed by a vegetation-free, snow-free, and ice-free
land surface, and energy loss via turbulent sensible (H), latent (LE), and ground (G) heat fluxes.
$R_N$ is the difference between incoming and outgoing longwave (LW) and shortwave (SW)
radiation where net LW is a function of atmospheric and surface temperatures, and net SW is a
function of incoming solar radiation and surface albedo. In terrestrial ecosystems G is therefore
modulated by vegetation function and structure, snow cover, topography, and hydrology (Smith,
1975; Betts & Ball, 1997; Eaton *et al.*, 2001; Zhang, 2005; Stiegler *et al.*, 2016; Helbig *et al.*,
2016a). Vegetation exerts strong controls on albedo, surface conductance, and surface
temperature (Betts & Ball, 1997; Betts *et al.*, 1999; Helbig *et al.*, 2016a), and consequently
partitioning of the surface energy balance into its component fluxes (Eugster *et al.*, 2000). These
energy balance controls vary diurnally, seasonally, and spatially across arctic and boreal
ecosystems (e.g. Beringer *et al.*, 2005), and are sensitive to natural and anthropogenic
disturbances (Helbig *et al.*, 2016b).

Though usually small compared to gross soil-atmospheric heat fluxes (H and LE), G is

critically important, because it is the transfer of heat between the ground surface and the active



layer and permafrost. G occurs primarily by thermal conduction, and is a function of the
temperature gradient between the ground surface and the permafrost table (Kane *et al.*, 2001; but
see Fan *et al.*, 2011), and the thermal conductivity ($K_T$) of the soil. Thus, variability in thermal
dynamics of active layer and permafrost soils are most generally controlled by factors
influencing: 1) the temperature gradient between the ground surface and permafrost at a given
depth, and 2) the $K_T$ of active layer and permafrost soil substrates (Figure 1). Ground surface
temperature ($T_{SG}$) is governed by energy dynamics of the atmosphere and overlying plant
canopies, and ground cover influences on albedo, H, and LE (Figure 1). $T_{SG}$ is different from the
land surface temperature ($T_{SL}$), a measure typically used to assess ecosystem-climate interactions
(e.g. Urban *et al.*, 2013), because $T_{SL}$ includes tall-statured overlying vegetation canopies,
whereas $T_{SG}$ includes only ground-cover vegetation (e.g. mosses and lichens), bare soil, or plant
litter that functionally represents the ground surface. Once energy has been absorbed at the
ground surface and $T_{SG}$ is elevated, soil $K_T$ will dictate how much of this energy is transferred
downward into the soil. Here we focus on $T_{SG}$ and $K_T$ because they are more dynamic than
permafrost temperature and will mediate permafrost responses to climate and associated carbon
cycle consequences, particularly in the coming decades to centuries.

**2.1 Vegetation canopies during the growing season**

Vegetation canopies attenuate incoming solar radiation (Juszak *et al.*, 2014; 2016),

thereby reducing radiation at the ground surface and subsequently $T_{SG}$. Canopy removal and
addition experiments illustrate that shrub canopies insulate tundra soils in summer, maintaining
soil temperatures upwards of 2°C cooler than adjacent tall shrub-free areas (Bewley *et al.*, 2007;
Blok *et al.*, 2010; Myers-Smith & Hik, 2013; Nauta *et al.*, 2014). Canopy shading has also been





linked to decreased soil temperatures in both evergreen (Jean & Payette, 2014a; 2014b; Roy-
Léveillée *et al.*, 2014; Fisher *et al.*, 2016) and deciduous (Iwahana *et al.*, 2005; Fedorov *et al.*,
2016) needleleaf boreal forests. Canopy removal experiments have resulted in substantial soil
warming, permafrost thaw and subsidence in ice-rich tundra (Blok *et al.*, 2010; Myers-Smith &
Hik, 2013; Nauta *et al.*, 2014) and deciduous needleleaf forests (Iwahana *et al.*, 2005; Fedorov *et*
*al.*, 2016). In the latter case, ecosystem recovery and winter processes lead to permafrost
stabilization in the decades after clearing (Fedorov *et al.*, 2016). Increases vegetation stature will
tend to decrease $T_{SG}$ and local soil cooling during the summer months when plant canopies are
present.

Whereas increases in tree and shrub cover reduce solar radiation at the ground surface,

the increased canopy stature and complexity generally reduces canopy albedo leading to an
overall increase of the canopy $R_N$ (Beringer *et al.*, 2005; Chapin *et al.*, 2005; Sturm *et al.*, 2005;
Loranty *et al.*, 2011). However, albedo may increase when shrubs replace bare ground or wet
tundra (Blok *et al.*, 2011b; Gamon *et al.*, 2012) or depending on changes in community
composition or structure (Williamson *et al.*, 2016). During the growing season these albedo
differences are relatively small (Juszak *et al.*, 2016), and associated changes in $R_N$ have not yet
been linked to soil thermal dynamics at the ecosystem scale (e.g. Beringer *et al.*, 2005).
Moreover, observations of lower $T_{SL}$ for boreal forest canopies relative to adjacent non-forested
lands due to higher LE flux (Li *et al.*, 2015) highlight the importance of canopy controls on
transpiration when considering how vegetation change affects land surface energy partitioning.
In summary, during the growing season there is no clear evidence for altered ecosystem scale G
associated with local evaporative cooling (Li *et al.*, 2015) or increased sensible heating as a



function of canopy albedo (Beringer *et al.*, 2005), likely because these effects are overwhelmed
by canopy light attenuation.

**2.2 Vegetation canopies during the non-growing season**

Snow covers much of the arctic and boreal regions for long periods each year and is a

critical driver of ground temperature (Goodrich, 1982; Stieglitz, 2003).   Deep and/or low-
density snow has low $K_T$ and therefore reduces heat flux from the ground to the atmosphere
during the non-growing season when air temperatures are typically colder than soil temperatures.
Snow depth is initially controlled by the timing and intensity of snowfall, but wind can
redistribute snow according to local topography, vegetation structure, landscape position and
wind direction, leading to high heterogeneity in snow cover and depth (Walker *et al.*, 2001;
Kershaw & McCulloch, 2007). Snow physical and insulative properties can also vary on the
scale of broad ecoregions as a result of differences in air temperature, wind, precipitation, and
vegetation cover (Sturm *et al.*, 1995). For example, high thermal conductivity and density of
snow in tundra relative to boreal ecosystems has been linked to differences in soil temperatures
(Gouttevin *et al.*, 2012; Mamet & Kershaw, 2013). Snow cover in the shoulder seasons (freeze-
up and thaw periods) can cool soils as a result of albedo effects, but generally ground insulation
from snow cover during the extended winter period dominates the snow effects on G. For
example, across the Alaskan arctic, ground surface temperatures are estimated to be 4°C to 9°C
warmer as a result of higher snow cover (Zhang, 2005).

In tundra, shrub canopies trap blowing snow, leading to localized deepening of snow

cover and higher winter soil temperatures (Sturm *et al.*, 2001; Liston *et al.*, 2002; Sturm *et al.*,
2005; Marsh *et al.*, 2010; Myers-Smith & Hik, 2013; Domine *et al.*, 2015). However, shrub





canopies can bend in winter under the snowpack leading to potentially different amounts of snow
trapping in years with heavy wet snow versus dry snow in early winter (Marsh *et al.*, 2010;
Ménard *et al.*, 2014). But even buried vegetation can lead to turbulent airflow that transports
snow into complex patterns (Filhol & Sturm, 2015) resulting in spatially variable ground
temperatures within a given year. In some cases vegetation-snow interactions can also have a
negative effect on winter ground temperature, leading to soil cooling. In northeast Siberia, large
graminoid tussocks exposed above the snowpack in early winter create gaps in the insulating
snow layer, which leads to lower ground temperatures, earlier active layer freezing and cooling
of surface permafrost (Kholodov *et al.*, 2012).

In the boreal forest, the presence of trees strongly reduces the wind regime and snow

redistribution typical of tundra (Baldocchi *et al.*, 2000). While there is less wind-distribution in
boreal forests than in the more open tundra, tree composition and density impact snow
distribution and depth through interception of snow by the canopy branches and subsequent
evaporation and sublimation. This results in lower snow inputs in dense forests and areas of
shallow snow underneath individual trees (Rasmus *et al.*, 2011). This winter effect of tree
density on snow cover may, in part, explain the negative relationship found between larch stand
density and ground thaw (Webb *et al.*, 2017) and is consistent with the effects of winter warming
experiments on summertime active layer dynamics (e.g. Natali *et al.*, 2011). However at treeline
or areas with patchy tree cover forests can trap blowing snow leading to elevated soil
temperatures in winter (Roy-Léveillée *et al.*, 2014)

Tall-statured vegetation canopies that protrude above the snowpack decrease land surface

albedo. While the accompanying increases in $R_N$ will lead to sensible heating of the atmosphere
at regional to local scales (Chapin *et al.*, 2005), they do not have a direct influence on $T_{SG}$ or $K_T$.



In the spring thaw period when snow covers the landscape and solar radiation is high, this
increase in $R_N$ is largest (Liston *et al.*, 2002; Pomeroy *et al.*, 2006; Marsh *et al.*, 2010) and may
accelerate snow melt (Sturm *et al.*, 2005; Loranty *et al.*, 2011). This could lead to a longer snow-
free season and greater G during the growing seaons, however, this snow-reducing effect can be
offset by the snow-trapping effects of vegetation (Sturm *et al.*, 2005). Changes in the length of
the snow-free season because of altered canopy albedo could lead to changes in G; however,
such an effect has not been observed. While canopy albedo does not directly influence G at the
ecosystem scale, regional climate feedbacks associated with albedo changes (described below)
may influence permafrost thermal dynamics (Lawrence & Swenson, 2011; Bonfils *et al.*, 2012).

**2.3 Groundcover impacts on ground surface temperature**
Ground cover in permafrost ecosystems may include bare soil, plant litter, lichens, and
mosses. Unlike vascular plant canopies, moss and lichen are in close thermal contact with the
underlying soil layers so heat can be transferred from the vegetation into the soil (and vice versa)
via conduction (e.g. O'Donnell *et al.*, 2009; Yi *et al.*, 2009). Differences in albedo and LE are the
primary causes of variability in $T_{SG}$ among ground cover types. Under moist conditions, non-
vascular evaporation rates are generally high, leading to surface cooling (Heijmans *et al.*, 2004a;
2004b). Under dry conditions taxonomic level differences in physiological responses to drought
(Heijmans *et al.*, 2004b), can lead to large differences in $T_{SG}$ (Stoy *et al.*, 2012). Increased LE
from bare soil after experimental- (Blok *et al.*, 2011a) and disturbance-induced (Rocha &
Shaver, 2011) moss removal illustrates the importance of non-vascular plant physiology, and
highlights the relatively high potential for evaporative cooling from bare soil surfaces. Low
hydraulic conductivity in mosses relative to organic and mineral soils may result in suppression



of LE once moisture held in surface vegetation is depleted, whereas higher hydraulic
conductivity in underlying soil layers may allow for evaporation of deeper soil moisture and
increased LE observed with moss removal (Rocha & Shaver, 2011; Blok *et al.*, 2011a). Albedo
differences between common moss and lichen species may also contribute to large differences in
$T_{SG}$; in ways that either amplify or ameliorate the effects of physiological differences in
evaporative cooling (Stoy *et al.*, 2012; Loranty *et al.*, 2018). Variability in ground cover can
correspond to large differences in $T_{SG}$ that depend on the joint effects of albedo and LE, and are
strongly dependent on available moisture. However the extent to which an increase in $T_{SG}$ leads
to an increase in G depends upon $K_T$ of the groundcover and soil layers.

**2.4 Impacts of ground cover and soil properties on thermal conductivity**

Soil $K_T$ , which often includes the moss layer where present, affects the rate of heat

transfer through the soil profile across a temperature gradient between the ground surface and the
soil at a given depth. $K_T$ varies throughout the soil profile with soil moisture and composition.
Under dry conditions, mosses have among the lowest $K_T$ , followed by organic and then mineral
soils (Hinzman *et al.*, 1991; O'Donnell *et al.*, 2009). Moss and organic soil layers have very low
$K_T$ owing to high porosity, and $K_T$ typically increases with soil bulk density (Hinzman *et al.*,
1991; O'Donnell *et al.*, 2009). Mineral soils typically have higher $K_T$ than organic soils (Kane *et*
*al.*, 1989; Hinzman *et al.*, 1991; Romanovsky & Osterkamp, 2000), and fine textured clay
mineral soils have lower $K_T$ than silt or sand (Johansen, 1977). In general, ecosystems with thick
moss and organic soil (e.g. peat) layers with low bulk density tend to have low G and shallow
active layers (Woo *et al.*, 2007; Fisher *et al.*, 2016).



Soil and moss moisture influences their thermal dynamics in a variety of important ways.
Linear increases in $K_T$ with moisture content (O'Donnell *et al.*, 2009; Soudzilovskaia *et al.*,
2013) have strong impacts on G, soil temperatures, and active layer dynamics. Under saturated
conditions, $K_T$ values of mineral soils remain higher than in organic soils and mosses (Hinzman
*et al.*, 1991; Romanovsky & Osterkamp, 2000; O'Donnell *et al.*, 2009), so the general pattern of
increasing $K_T$ with depth/bulk density is maintained. Local- and ecosystem-scale observations of
warmer soil temperatures and deeper thaw depths in areas of perennially elevated soil moisture
(Hinkel *et al.*, 2001; Hinkel & Nelson, 2003; e.g. Shiklomanov *et al.*, 2010; Curasi *et al.*, 2016)
indicate increases in $K_T$ outweigh the concurrent increase in specific heat capacity associated
with increasing moisture content. Similarly, interannual variability in soil moisture and active
layer thickness are positively related across a range of spatial scales (Iijima *et al.*, 2010; Park *et*
*al.*, 2013).
Liquid water and water vapor can also warm soils through non-conductive heat transfer
(Hinkel & Outcalt, 1994; i.e. water movement; Kane *et al.*, 2001). Here, the timing and source of
water is important. For example, infiltration of snowmelt in spring does not deliver substantial
heat to the soil because the water temperature is very close to freezing (Hinkel *et al.*, 2001) and
the near-surface soil horizons are mostly frozen. Alternatively, condensation of water vapor in
frozen soils can lead to fairly rapid temperature increases during spring melt (Hinkel & Outcalt,
1994). Heat delivery from groundwater flow has been implicated as a cause for permafrost
degradation in areas of discontinuous permafrost in interior Alaska (Jorgenson *et al.*, 2010). The
hydraulic properties of soil horizons are especially important in this regard. Unsaturated peat and
organic-soil horizons with large interconnected pore spaces generally promote non-conductive
transport of heat in soils unless the substrate is dry enough that it absorbs water.





The relative importance of non- conductive heat transfer on permafrost thermal
dynamics is difficult to determine. Observations of elevated soil temperature, active layer
thickness, and thermal erosion in areas with poorly drained or inundated soils (Woo, 1990; e.g.
Jorgenson *et al.*, 2010; Curasi *et al.*, 2016) suggest the effects of soil moisture on $K_T$ may have
stronger influences than convective processes on soil thermal dynamics. However, several recent
studies indicate that heat advected in groundwater may promote permafrost thaw (de Grandpré *et*
*al.*, 2012; Sjöberg *et al.*, 2016). Soil moisture distribution within the soil profile is important as
well; dry surface organic layers with low $K_T$ may buffer against warmer air temperatures even
though deeper soils may have high $K_T$ associated with moisture and soil composition (e.g. Rocha
& Shaver, 2011). Observations of co-varying heterogeneity in soil structure, temperature, and
moisture also illustrate the importance of spatio-temporal variability in soil moisture and $K_T$ for
understanding permafrost soil thermal dynamics (Boike *et al.*, 1998).
In wet soils the large latent heat content of soil moisture can delay freezing of the active
layer (i.e. extend the freeze-up duration; Romanovsky & Osterkamp 2000). The period during
which soil active layer temperatures remain constant near 0°C as latent heat is released form soil
moisture is commonly referred to as the 'zero-curtain' (Outcalt *et al.*, 1990).  Longer zero-
curtain periods promote warmer winter active layer and permafrost temperatures (Outcalt *et al.*,
1990; Morse *et al.*, 2015). Soil thaw during spring tends to occur more rapidly than freeze-up
during autumn, despite the high latent heat required to thaw ground ice, likely due to increases in
$K_T$ associated with snowmelt infiltration and/or latent heat released by condensation of water
vapor (Hinkel & Outcalt, 1994). Excess ground ice deeper in the active layer or permafrost
requires larger amounts of latent heat energy to melt, and so typically buffer permafrost soils
against thaw (Halsey *et al.*, 1995). However, when this type of ground ice does melt, it can lead



to an array of physical and ecological changes via thermokarst development (Mamet *et al.*,
2017), which further alter the soil thermal regime and can promote further warming (Osterkamp
*et al.*, 2009; Kokelj & Jorgenson, 2013).

**2.5 Interacting ecosystem influences on ground heat flux**

The mechanisms described in the previous sections are relatively well understood

individually, but when considered in concert, the relative importance of specific processes is
often unclear. This is particularly true when ecological processes co-vary, or have opposing
effects on permafrost soil thermal dynamics. For example, concurrent accumulation of organic
soil and canopy leaf area make it difficult to quantify the relative importance of each when
considering differences in active layer properties across successional gradients (Jorgenson *et al.*,
2010). Consequently, the magnitude of permafrost soil temperature responses to ecological
change is uncertain.

Though there are a number of studies that have examined the role of variation in

vegetation canopy cover, soil moisture, and ground/soil thermal properties on the permafrost
thermal regime, few have fully isolated the relative contribution of each process to variation in
active layer thickness or soil temperatures (Jiang *et al.*, 2015). For example, in addition to
increasing radiation at the ground surface, canopy removal experiments (Blok *et al.*, 2010; e.g.
Fedorov *et al.*, 2016) may also elevate soil moisture via reductions in plant water use. In a recent
study by Fisher et al. (2016) examining the impact of multiple processes on active layer
thickness in Canadian boreal forest overstory leaf area to be most important, followed by moss
thickness and understory leaf area. Further, this study revealed that moisture in deeper soil layers
modified the impacts of vegetation whereas surface soil moisture did not (Fisher *et al.*, 2016).



Ecosystem influences on moisture distribution throughout the soil profile, particularly in relation
to evapotranspiration, are not well characterized and will likely become increasingly important
with continued climate warming (Swann *et al.*, 2010).

It is also important to consider the relative contributions of seasonal variation in

ecosystem influences on permafrost thermal dynamics, and the potential for temporal
autocorrelation at annual timescales. Myers-Smith and Hik (2013) found that winter warming
associated with snow-trapping by shrub canopies elevated soil temperatures by 4-5 °C whereas
canopy shading led to 2 °C cooling in summer. Similarly, relative to non-forested palsas,
forested palsas in eastern Canada exhibited winter soil warming associated with snow trapping
but slower rates of permafrost thaw due to summer cooling associated with thicker organic layers
and canopy shading (Jean & Payette, 2014a; 2014b). Additionally, these studies observed
delayed freeze-up and later spring thaw associated with late fall precipitation that resulted in
complex relationships between annual air and soil temperatures and active layer depths (Jean &
Payette, 2014b). The magnitude of these effects likely varies spatially with patch size and
climatic controls, making it difficult to distinguish the relative importance of summer versus
winter processes, as well as potential links across successive growing seasons.

Disentangling the relative impacts of multiple ecosystem characteristics on G will

become increasingly important as ecological responses to continued climate warming may lead
to shifts in ecosystem distribution (Pearson *et al.*, 2013; Abbott *et al.*, 2016), potentially resulting
in novel ecosystems with no current eco-climatic analogs (Macias-Fauria *et al.*, 2012). Because
ecosystems influence permafrost soil thermal dynamics in a variety of ways, such shifts in
ecosystem distribution are likely to fundamentally alter rates of permafrost thaw with projected
future warming. This will occur directly via altered ecosystem surface energy dynamics that



affect G and indirectly through changes to the surface energy balance that feed back to climate
(e.g. Figure 1). The following sections describe ongoing and anticipated ecosystem responses to
climate and associated changes to G via impacts on $T_{SG}$ or $K_S$, and then the associated regional to
global scale atmospheric feedbacks.

**3 Ecosystem change with implications for permafrost thermal dynamics**
Vegetation productivity and community composition are changing in response to longer
and warmer growing seasons associated with amplified climate warming across the Arctic.
Relationships between air temperature and soil thermal dynamics vary with ecosystem properties
and will therefore evolve as ecosystems respond to climate change. Ecosystem structural and
functional characteristics that influence soil thermal dynamics may be altered directly by
ecosystem responses to climate change, or indirectly by climatic alteration of disturbance
processes that in turn modify ecosystems (e.g. O'Donnell *et al.*, 2011a). In this section, we
outline key ecosystem changes arising from direct and indirect climate responses (summarized in
Figure 2), and describe how these changes are likely to affect permafrost soil thermal dynamics
via impacts on processes described above.

**3.1 Vegetation change in response to climate**
In tundra ecosystems, increases in vegetation productivity inferred from satellite
observations (Jia *et al.*, 2003; Beck & Goetz, 2011) have been linked to shrub expansion and
accelerated annual growth at locations throughout the Arctic (Tape *et al.*, 2006; Forbes *et al.*,
2010; Macias-Fauria *et al.*, 2012; Frost & Epstein, 2014). However, warming experiments
indicate that productivity increases may occur without shifts in the dominant vegetation type



393 (Walker *et al.*, 2006; Elmendorf *et al.*, 2012b), and dendroecological observations illustrate that

394 shrub responses to temperature are moderated by moisture and nutrient availability and are

395 highly heterogeneous in space and time (Zamin & Grogan, 2012; Myers-Smith *et al.*, 2015;

396 Ackerman *et al.*, 2017). Despite the high degree of heterogeneity in tundra vegetation responses

397 to warming (Elmendorf *et al.*, 2012a), there are several consistent changes that include increased

398 vegetation height, increased litter production, decreased moss cover (Elmendorf *et al.*, 2012b),

399 and increased graminoid cover in lowland permafrost features (Malmer *et al.*, 2005; Johansson *et*

400 *al.*, 2006; Malhotra & Roulet, 2015). However, reductions in greenness in some regions (referred

401 to as 'browning') driven by, for example, reduced summer warmth index (Bhatt *et al.*, 2013) or

402 acute 'browning events' from disturbances such as winter frost droughts (Bjerke *et al.*, 2014;

403 Phoenix & Bjerke, 2016) add complexity to predicting vegetation change and hence subsequent

404 impacts on permafrost.

405  Enhanced tundra vegetation productivity may reduce summer soil temperatures via

406 ground shading and increase winter soil temperatures via effects on snow depth and density. The

407 effect of declining moss cover will depend on the balance between reduced insulation (i.e. $K_T$)

408 and latent cooling associated with increased soil evaporation. Vegetation change may also alter

409 organic soil accumulation rates via altered litter quality and quantity (Cornelissen *et al.*, 2007).

410 This overall effect on soil $K_T$ will depend on the net effects of changing litter inputs, lability, and

411 decomposition rates with warming (Hobbie, 1996; Hobbie & Gough, 2004; Cornelissen *et al.*,

412 2007; Christiansen *et al.*, 2018; Lynch *et al.*, 2018).

413  Belowground vegetation dynamics are more difficult to study, but recent observations

414 indicate that the below ground growing season length (period of unfrozen temperatures allowing

415 for plant growth) can be greater than that aboveground (Blume-Werry *et al.*, 2015; Radville *et*



*al.*, 2016). These differences likely vary with depth due to effects related to the progression of
soil freezing and thawing (Rydén & Kostov, 1980). Thus, rooting depth and lateral root
distributions will influence the below-ground phenology differentially for deep-rooted (e.g.,
sedge) versus shallow-rooted (e.g., shrub) species (Bardgett *et al.*, 2014; Iversen *et al.*, 2015),
which may alter soil moisture via plant water uptake under future warming related vegetation
change increased active layer depth. The changing above- and below-ground growth phenology
of tundra plants (Blume-Werry *et al.*, 2015; Iversen *et al.*, 2015; Radville *et al.*, 2016) could also
favor the proliferation of certain functional groups or species creating potential feedbacks to
vegetation change. In addition to belowground phenology, total root production could also
increase in response to warming (e.g. Xue *et al.*, 2015). However, increased nutrient availability
from warming could decrease root production relative to aboveground production (Keuper *et al.*,
2012; Poorter *et al.*, 2012). The net effect of climate change induced belowground changes on
soil thermodynamics is unclear.

Boreal forest responses to climate in recent decades were generally more heterogeneous

than those observed in tundra ecosystems due to a variety of interacting factors including species
differences in physiology, disturbance regimes, and successional dynamics. Initial satellite
observations of boreal forest productivity increases (Myneni *et al.*, 1997) have slowed or even
reversed in recent decades (Beck & Goetz, 2011; Guay *et al.*, 2014). Tree ring analyses confirm
productivity declines associated with temperature induced drought stress in interior Alaska
boreal forests (Barber *et al.*, 2000; Walker & Johnstone, 2014; Juday *et al.*, 2015; Walker *et al.*,
2015), and have been used to corroborate satellite observations (Beck *et al.*, 2011). Similarly,
drought-induced mortality has been observed at the southern margins of Canadian boreal forests
(Peng *et al.*, 2011) where correspondence between satellite and tree ring records have also been



observed (Berner *et al.*, 2011). In Siberia, positive forest responses to air temperatures observed
in tree rings and satellite observations near latitudinal tree lines give way to declines in tree
growth further south (Lloyd *et al.*, 2010; Berner *et al.*, 2013). These results are in line with
ecosystem-scale observations of suppressed transpiration under high vapor pressure deficits and
low soil moisture conditions (Lopez C *et al.*, 2007; Kropp *et al.*, 2017). More generally, forests
growing on continuous permafrost exhibit more widespread productivity increases (Loranty *et*
*al.*, 2016), suggesting that permafrost may buffer against drought stress. However, waterlogged
soil resulting from permafrost thaw can also lead to unstable soils and forest mortality (Baltzer *et*
*al.*, 2014; Iijima *et al.*, 2014; Helbig *et al.*, 2016a).

The extent to which ongoing boreal forest productivity changes influence permafrost soil

thermal dynamics is not entirely clear. If forest canopy cover changes with productivity (e.g.
canopy infilling or increased leaf area), then changes in ground shading could alter ground
thermal regimes. Increases in forest cover have been observed in northern Siberia (Frost &
Epstein, 2014); however, it is unclear whether the cause is climate warming or ecosystem
recovery after fire. Conversely, productivity declines are more pronounced in high-density
forests (Bunn & Goetz, 2006) and, consequently, browning trends associated with mortality in
southern boreal forests (Peng *et al.*, 2011) may increase radiation at the ground surface.
Additionally, if browning is indicative of drought stress, vegetation may enhance the insulation
of organic soils by further depleting of soil moisture via plant water uptake (Fisher *et al.*, 2016).
Forest mortality and declines in canopy cover in southern boreal forests as a consequence of
permafrost thaw (Helbig *et al.*, 2016a) may feedback positively to permafrost thaw. A clearer
understanding of boreal forest structural and ecohydrological changes associated with
widespread productivity changes is necessary.



### 3.2 Wildfire disturbance


Wildfire is the dominant disturbance in the boreal forest and is increasingly present in

arctic tundra. Wildfire influences surface energy dynamics via impacts on vegetation and surface
soil properties, likely accelerating permafrost thaw (Burn, 1998; Viereck *et al.*, 2008; O'Donnell
*et al.*, 2011a; Jafarov *et al.*, 2013; Brown *et al.*, 2015; Jones *et al.*, 2015).Vegetation combustion
and mortality increases radiation at the ground surface. The combustion and charring of moss
and organic soil lowers albedo and increases $K_t$, leading to warmer soils with deeper active
layers in the decades following a fire. (Yoshikawa *et al.*, 2003; Liljedahl *et al.*, 2007; Rocha &
Shaver, 2011; French *et al.*, 2016). In boreal forests, loss of canopy cover increases albedo
during the snow-covered period (Jin *et al.*, 2002; Lyons *et al.*, 2008; Jin *et al.*, 2012), which may
result in local atmospheric cooling (Lee *et al.*, 2011). However, such atmospheric cooling has not
been linked to soil climate, and canopy loss may also result in a deeper snowpack, which inhibits
ground cooling during winter (Kershaw, 2001). In general, wildfire effects on permafrost soil
climate are primarily the result of altered growing season surface energy dynamics.

The magnitude of wildfire effects on soil temperature is closely linked to burn severity,

as indicated by the degree of organic soil combustion and the post-fire organic horizon thickness
(Kasischke & Johnstone, 2005). Post-fire recovery of the organic-soil horizon can allow recovery
of soil temperature and active layer thickness to pre-fire conditions (Rocha *et al.*, 2012).
However, relatively warm discontinuous zone permafrost is often ecosystem-protected by
vegetation and organic horizons (Shur & Jorgenson, 2007), thus loss or reduction of organic soil
may result in the irreversible thaw or loss of permafrost (Romanovsky *et al.*, 2010; Jiang *et al.*,
2015). Site-based model simulations suggest that fire-driven change in organic-horizon thickness



is the most important factor driving post-fire soil temperature and permafrost dynamics (Jiang *et*
*al.*, 2015).
Wildfire impacts on permafrost also vary spatially with ecosystems and topography. For
instance south-facing forest stands tend to burn more severely than north-facing stands (Kane *et*
*al.*, 2007). Further, poorly drained toe-slopes burn less severely than more moderately drained
upslope landscapes. These topographic effects on burn severity can strongly influence the
response of soil temperature and permafrost to fire (O'Donnell *et al.*, 2009). The loss of
transpiration due to the combustion of trees may result in wetter soils in recently burned stands
compared to unburned stands (O'Donnell *et al.*, 2011a). However, other studies have
documented drier soils in burned relative to unburned stands (Jorgenson *et al.*, 2013),
particularly at sites underlain by coarse-grained, hydrologically conductive soils. Post-fire
thawing of permafrost can increase the hydraulic conductivity of mineral soils due to ice loss,
leading to enhanced infiltration of soil water and soil drainage. Post-fire changes in soil moisture
and drainage can function as either a positive or negative feedback to permafrost thaw
(O'Donnell *et al.*, 2011b). Recent evidence also indicates that mineral soil texture is an important
control on post-fire permafrost dynamics (Nossov *et al.*, 2013).
While the magnitude of fire effects on G and active layer depth is typically governed by
burn severity, the persistence of these changes depends on ecosystem recovery (Jorgenson *et al.*,
2013). Albedo returns to pre-fire levels within several years after fire (Jin *et al.*, 2012) due to
fairly rapid recovery of vegetation (Mack *et al.*, 2008). Recovery of moss and re-accumulation of
the organic-soil horizon further facilitate recovery of soil temperatures and permafrost, and may
occur within several decades (e.g. Loranty *et al.*, 2014b). Finally, recovery of vegetation
canopies over decades to centuries gradually reduces incident radiation at the ground surface to



pre-fire levels. The effects of fire on $T_{SG}$ and permafrost are well understood, and it may be
reasonable to expect similar effects in the future that are amplified as fire exposes permafrost
soils to increasingly warmer atmospheric temperatures. However, changes in the severity and
extent of wildfires can result in new ecosystem dynamics with implications for permafrost that
do not confer linearly from current eco-climatic conditions.

Recent warming at high latitudes has increased the spatial extent, frequency, and severity

of wildfires in North America (Turetsky *et al.*, 2011; Rocha *et al.*, 2012) to levels that are
unprecedented in recent millennia (Hu *et al.*, 2010; Kelly *et al.*, 2013). Fire regimes in boreal
forests in Eurasia remain poorly characterized (Kukavskaya *et al.*, 2012), though several studies
indicate that fire extent and frequency are likely increasing with climate warming (Kharuk *et al.*,
2008; 2013; Ponomarev *et al.*, 2016). Recovery of soil thermal regimes and permafrost after fire
is strongly influenced by ecosystem recovery, and recent studies have established links between
burn severity and post-fire succession (Johnstone *et al.*, 2010; Alexander *et al.*, 2018).
Consequently, burn severity is likely the dominant factor controlling the effects of wildfire on
permafrost soil thermal dynamics.

In boreal North America, low-severity fires in upland black spruce forest typically foster

self-replacing post-fire vegetation trajectories while high-burn severity fosters a transition to
deciduous dominated forests. (Johnstone *et al.*, 2010). In addition to changes in canopy effects
on ground shading, this transition also leads to reductions in post-fire accumulation of the soil
organic layer (Alexander & Mack, 2015). Observations of mean annual soil temperatures that are
1-2 °C colder in soils underlying black spruce forests compared to deciduous forests (Jorgenson
*et al.*, 2010; Fisher *et al.*, 2016) indicate that burn severity influences on post-fire succession will
lead to alternate soil temperature and permafrost recovery pathways as well.



In Siberian larch forests, post-fire recovery is impacted by fire severity and seed dispersal
(Figure 3). High burn severity fires promote high rates of seedling recruitment and subsequent
forest stand density (Sofronov & Volokitina, 2010; Alexander *et al.*, 2018) when dispersal is not
limited. But since larch are not serotinous and seed rain varies from year to year, high burn
severity does not guarantee succession to high-density forests. Recovery tends to be slow and
highly variable (Berner *et al.*, 2012; Alexander *et al.*, 2012b). Wide ranges of post-fire moss
accumulation and forest regrowth have been observed, though consequences for permafrost are
unclear (Furayev *et al.*, 2001). Observed declines in permafrost thaw depth with increasing
canopy cover (Webb *et al.*, 2017) support the notion of a link between fire severity and
permafrost soil thermal dynamics. However, the combined effects of fire and climatic warming
and drying could lead to widespread conversion of larch forests to steppe (Tchebakova *et al.*,
2009), whereas declines in fire could result in increased cover of evergreen needleleaf species
(Schulze *et al.*, 2012). Thus the impacts of fire on permafrost in Siberia remain uncertain.
In tundra ecosystems fire is becoming increasingly common (Rocha *et al.*, 2012). Fire-
induced transitions from graminoid- to shrub-dominated ecosystems have been observed in
several instances (Landhäusser & Wein, 1993; Racine *et al.*, 2004; Jones *et al.*, 2013), while in
others recovery of graminoid-dominated ecosystems has occurred (Vavrek *et al.*, 1999; Barrett *et*
*al.*, 2012; Loranty *et al.*, 2014b). If unusually large tundra fires with high burn severity (e.g.
Jones *et al.*, 2009) occur more regularly fire induced transitions from graminoid to shrub tundra
may become more common (Jones *et al.*, 2013; Lantz *et al.*, 2013). A shift to shrub dominance
could buffer permafrost soils from continued climate warming during summer (e.g Blok *et al.*,
2010; Myers-Smith & Hik, 2013) or promote warmer soils in winter (Lantz *et al.*, 2013; Myers-
Smith & Hik, 2013) at the ecosystem-scale depending on how topography and the spatial



distribution of shrubs impact snow redistribution (Essery & Pomeroy, 2004; Ménard *et al.*,
2014), In addition, there is evidence that thermal erosion as a consequence of fire may facilitate
shrub transitions, especially in areas of ice-rich permafrost (Bret-Harte *et al.*, 2013; Jones *et al.*,
2013), and the associated changes in local hydrology and topography will also impact soil
temperature dynamics.

**3.3 Permafrost thaw, thermokarst disturbance, and hydrologic change**
Permafrost thaw can occur in two primary modes, as determined by pre-thaw ground ice
content. In terrain underlain by low ground ice content (typically < 20% by volume), the soil
profile can thaw from the top down without disturbing the surface in what is termed thaw-stable
permafrost degradation (Jorgenson *et al.*, 2001). Alternatively, in ice-rich terrain, when ground
ice volume exceeds unfrozen soil pore space (usually > 60%), permafrost thaw causes surface
subsidence or collapse, termed thermokarst (Kokelj & Jorgenson, 2013). Thermokarst is the
predominant disturbance in arctic tundra and is an important disturbance in boreal forests
underlain by permafrost (Lara *et al.*, 2016). Recent evidence indicates increasing prevalence of
thermokarst features during the last half-century (Jorgenson *et al.*, 2006; 2013; Liljedahl *et al.*,
2016; Mamet *et al.*, 2017), though circum-arctic prevalence and change of thermokarst extent are
poorly constrained (Yoshikawa & Hinzman, 2003; Lantz & Kokelj, 2008; Olefeldt *et al.*, 2016).
Thermokarst features form over the course of weeks to decades, can involve centimeters to
meters of ground surface displacement, and typically lead to dramatic changes in ecosystem
vegetation and soil properties (e.g. Osterkamp *et al.*, 2000; Douglas *et al.*, 2016). Ecological
responses to thermokarst formation can act as either positive or negative feedbacks to continued
thaw, depending on how thermokarst formation affects vegetation and hydrology, including



snow cover (Kokelj & Jorgenson, 2013). Thermokarst could affect 20–50% of the permafrost
zone by the end of the century, according to projections of permafrost degradation and the
distribution of ground ice (Zhang *et al.*, 2000; Slater & Lawrence, 2013; Abbott & Jones, 2015).
Upland thermokarst in the discontinuous permafrost zone already impacts 12% of the overall
landscape in some areas and up to 35% of some vegetation classes (Belshe *et al.*, 2013).
Following initial thaw, hydrologic conditions play an important role in the subsequent
evolution of thermokarst features because the high thermal conductivity of water can increase
heat flux to the active layer and permafrost (Nauta *et al.*, 2015). Lowland and upland thermokarst
may have contrasting effects on surface hydrology, with lowland thermokarst initially increasing
wetness (e.g. O'Donnell *et al.*, 2012), but eventually leading to greater drainage if permafrost is
completely degraded (Anthony *et al.*, 2014). Upland thermokarst can either increase or decrease
surface wetness, depending on soil conditions and local topography (Abbott *et al.*, 2015; Abbott
& Jones, 2015; Mu *et al.*, 2017). Redistribution of water to thermokarst pits and gullies can lead
to drying in adjacent areas that have not subsided (Osterkamp *et al.*, 2009). In winter, increases
in snow accumulation in thermokarst depressions insulates soils (Stieglitz, 2003).
Thermokarst impacts vegetation and soils in a variety of ways. Active layer detachments
in uplands remove vegetation and organic soil, increasing energy inputs to deeper soil layers. In
upland tundra, shifts from graminoid- to shrub-dominated vegetation communities have been
observed with thaw, though communities varied locally with microtopography created by
thermokarst features themselves (Schuur *et al.*, 2007). In boreal forests, thermokarst and
permafrost thaw can cause transitions to wetlands or aquatic ecosystems (Jorgenson &
Osterkamp, 2005); whereas, vegetation community shifts are more subtle in uplands (Jorgenson
*et al.*, 2013). Permafrost thaw may also lead to a more nutrient rich environment (Keuper *et al.*,



2012; Harms *et al.*, 2014), but this depends on local soil properties. The succession of aquatic or
terrestrial vegetation can curb thaw through negative feedbacks and aggrade permafrost (Briggs
*et al.*, 2014).

**3.4 Zoogenic disturbance**

A large portion of the circumpolar Arctic is grazed by reindeer and caribou (both

*Rangifer tarandus* L.), and their grazing and trampling causes important long-term vegetation
shifts, namely inhibition of shrub proliferation (Olofsson *et al.*, 2004; Forbes & Kumpula, 2009;
Olofsson *et al.*, 2009; Plante *et al.*, 2014; Väisänen *et al.*, 2014). Besides direct consumption of
lichen and green biomass, large semi-domestic reindeer herds of northwest Eurasia also exert a
variety of impacts on biotic and abiotic components of Arctic and sub-Arctic tundra ecosystems
that have implications for permafrost thermal dynamics. For example, as reindeer reduce vertical
structure of vascular and nonvascular vegetation, they tend to decrease albedo (Beest *et al.*,
2016) and reduce thermal conductivity at the ground level (Olofsson, 2006; Fauria *et al.*, 2008),
which can lead to warmer soils (Olofsson *et al.*, 2001; van der Wal *et al.*, 2001; Olofsson *et al.*,
2004). Recent research has revealed that the consequences of climate warming on tundra carbon
balance are determined by reindeer grazing history (Zimov *et al.*, 2012; Väisänen *et al.*, 2014).
Historic and future grazing and trampling impacts on vegetation communities and soils will
continue to be important for understanding permafrost soil temperature responses to climate.

**3.5 Anthropogenic disturbance**

The most extensive direct anthropogenic disturbances within the permafrost zone occur

in three regions that have experienced widespread hydrocarbon exploration and extraction



activities: the North Slope of Alaska, the Mackenzie River Delta in Canada, and northwest
Russia, including the Nenets and Yamal-Nenets Autonomous Okrugs. The types of terrestrial
degradation commonly associated with the petroleum industry have historically included rutting
from tracked vehicles; seismic survey trails; pipelines, drilling pads and roads and the excavation
of the gravel and sand quarries necessary for their construction (Walker *et al.*, 1987; Huntington
*et al.*, 2013). A single pass of a vehicle over thawed ground can create ruts with increased $K_T$ due
to increased bulk density and soil moisture, while altered local hydrology can drain downslope
wetlands and, in both cases, lead to vegetation changes that persist for decades (Forbes, 1993;
1998). As a result of these combined factors, the increase from scale of impact to scale of
response can be several orders of magnitude (Forbes *et al.*, 2001). It has also been demonstrated
that even relatively small-scale, low intensity disturbances in winter, like seismic surveys over
snow-covered terrain, reduce microtopography, and increase ground temperatures and active
layer thaw depths (Crampton, 1977).

More recently, gravel roads and pads have become common, however this elevated

infrastructure causes other unanticipated impacts to the permafrost from accumulated dust, snow
drifts, and roadside flooding (Walker & Everett, 1987; 1991; Auerbach *et al.*, 1997; Raynolds *et
al.*, 2014). Over time, the warmer environments adjacent to roads have led to strips of earlier
phenology and shrub vegetation and even trees along both sides of most roads and buried
pipeline berms in the Low Arctic (Gill *et al.*, 2014). Aeolian sand and dust associated with gravel
roads or quarries can affect tundra vegetation and soils up to 1 km from the point source (Forbes,
1995; Myers-Smith *et al.*, 2006). At present, there is a concern that climate warming and
infrastructure are combining to enhance melting of the top surface of ice-wedges, leading to



more extensive ice-wedge thermokarst (Raynolds *et al.*, 2014; Liljedahl *et al.*, 2016)  and
cryogenic landslides  (Leibman *et al.*, 2014) in areas of intensive development.

**4 Local versus regional ecosystem feedbacks on permafrost thermal dynamics**
Interactions between ecosystem scale microclimate feedbacks and regional or global
climate feedbacks stemming from ecological change are complex and represent a key source of
uncertainty related to understanding permafrost soil responses to continued climate warming. If
changing ecosystem characteristics influencing permafrost thermal dynamics described above
are widespread, the accompanying changes in land surface water and energy exchange will feed
back to influence regional climate, and changes in greenhouse gas dynamics will feed back on
global climate (Chapin *et al.*, 2000b). Therefore, ecosystem changes that alter local permafrost
soil thermal dynamics may also lead to regional and global climate feedbacks that compound or
offset ecosystem-scale effects (Figure 4).

**4.1 Regional biogeochemical climate feedbacks**
The net biogeochemical climate effects of ecosystem change across the permafrost
regions will be a balance of changes in $CO_2$ uptake that accompany shifts in vegetation, and
changes in $CO_2$ and $CH_4$ release associated with shifts in autotrophic and heterotrophic
respiration, and fire and thermokarst disturbance. These feedback effects will be global in extent
and will not contribute directly to regional variability in permafrost thaw because greenhouse
gasses are well mixed in the atmosphere. Changes in the net $CO_2$ balance remain uncertain, but a
recent expert survey suggests that over the next century increases in vegetation productivity may
not be large enough to offset increases in carbon release to the atmosphere (Abbott *et al.*, 2016).



In tundra ecosystems, this conclusion is in line with projections of future biomass distribution
(Pearson *et al.*, 2013) and atmospheric inversions showing that increased autumn $CO_2$ efflux
offsets increases in uptake during the growing season (Welp *et al.*, 2016; Commane *et al.*, 2017).
In boreal forests, carbon cycle changes are more complex; long-term trends in the annual
amplitude of atmospheric $CO_2$ concentrations (Graven *et al.*, 2013; Forkel *et al.*, 2016) suggest
increases in biological activity while satellite observations and tree ring analyses suggest
widespread declines in productivity (Beck *et al.*, 2011). Further, model analyses indicate a
weakening terrestrial carbon sink associated with declining uptake, increases in respiration, and
disturbance (Hayes *et al.*, 2011), which is crucially important in boreal forests (Bond-Lamberty
*et al.*, 2013).

The net $CO_2$ effect of wildfire has typically been considered to be close to zero for

evergreen needleleaf forests in interior Alaska over historic fire return intervals (Randerson *et*
*al.*, 2006). However, the combined effects of climate warming and fire tend to reduce ecosystem
carbon storage by thawing permafrost (Harden *et al.*, 2000; O'Donnell *et al.*, 2011b; Douglas *et*
*al.*, 2014). Model simulations that include permafrost dynamics indicate ecosystem carbon losses
may become larger in the future with continued warming and intensification of the fire regime,
particularly for dry upland sites (Genet *et al.*, 2013; Jafarov *et al.*, 2013). These studies do not
account for potential changes in post-fire vegetation communities (Alexander *et al.*, 2012a)
however, the net effects of vegetation shifts on ecosystem carbon storage appear to be minimal
(Alexander & Mack, 2015). In tundra ecosystems larger and more severe fires lead to large soil
C losses (Mack *et al.*, 2011) that may be sustained over time due to permafrost thaw (Jones *et al.*,
2013; 2015). Across the permafrost region, available evidence suggests that fire will likely lead
to net carbon losses in the coming decades to centuries, thus acting as a positive feedback to





climate warming with associated effects on permafrost soils. The biophysical climate feedbacks
associated with fire are more immediate and will be stronger than the carbon cycle feedbacks
(Randerson *et al.*, 2006).

The effects of thermokarst on greenhouse gas dynamics depend largely on associated

hydrological changes. With increased drainage and surface drying, increased oxidation rates
reduce carbon accumulation (Robinson & Moore, 2000) and enhance $CO_2$ release (Frolking *et*
*al.*, 2006), and reduce $CH_4$ production (Abbott & Jones, 2015). When ground thaw is associated
with increased soil saturation, $CH_4$ production and emissions are increased (Johansson *et al.*,
2006; Olefeldt *et al.*, 2012; Abbott & Jones, 2015; Malhotra & Roulet, 2015; Natali *et al.*, 2015),
which can shift tundra from a net $CH_4$ sink (Jorgensen *et al.*, 2015) into a $CH_4$ source (Nauta *et*
*al.*, 2015). Thermokarst may also increase lateral transport of soil organic matter, which can
decrease $CO_2$ release (Abbott & Jones, 2015) and alter carbon processing downslope.
Thermokarst lakes emit $CH_4$, particulary along actively thawing lake margins (Walter *et al.*,
2007; 2008), and $CO_2$ (Kling *et al.*, 1991; Algesten *et al.*, 2004). However at millennial
timescales, thermokarst lakes can sequester carbon as lake sediments and peat accumulate (Jones
*et al.*, 2012; Anthony *et al.*, 2014). Currently thermokarst landscapes comprise upwards of 20%
of the permafrost region (Olefeldt *et al.*, 2016), however  their current and future impacts on the
global carbon balance remain poorly constrained.

**4.2 Regional biophysical climate feedbacks**

The biophysical effects of ecosystem change arising from shifts in surface energy

partitioning have climate feedback effects at scales ranging from local to regional and global.
Whereas biogeochemical climate feedbacks will influence global temperature in conjunction



with many other carbon cycle processes, biophysical feedbacks operating at local and regional
scales are likely to influence the spatial and temporal patterns of permafrost thaw with continued
warming. As described in the previous sections, changes in vegetation composition and structure
alter soil thermal dynamics via changes in G during the snow-free season (Chapin *et al.*, 2000a;
Beringer *et al.*, 2005). However, changes in G associated with vegetation change will also be
accompanied by changes in H and LE that may feedback to G, depending upon the scale of
impact.

Decadal ecosystem responses to climate inferred from 'greening' or 'browning' trends

are the most spatially pervasive change affecting vegetation in the permafrost zone (Loranty *et*
*al.*, 2016). Increases in leaf area and/or vegetation stature will generally reduce albedo, and these
effects are particularly pronounced during the spring and fall if enhanced productivity leads to
increased snow-masking by vegetation (Sturm *et al.*, 2005; Loranty *et al.*, 2014a). Reductions in
albedo will lead to sensible heating of the atmosphere (Chapin *et al.*, 2005) that may counteract
the effects of canopy shading on G, if albedo reduction occurs at sufficiently large spatial scales
(Lawrence & Swenson, 2011; Bonfils *et al.*, 2012). The magnitude and spatial extent of height
increases are crucial to determine the net feedback strength, but these quantities remain largely
unknown.

A second important but relatively unexplored feedback relates to evaporative cooling of

the land surface associated with increases in LE (but see Swann *et al.*, 2010). Productivity
increases are likely accompanied by increases in evapotranspiration (Zhang *et al.*, 2009), which
have been shown to mitigate temperature increases at global scales by increased cloud cover,
which may reduce incoming short-wave radiation reaching the Earth's surface (Zeng *et al.*,
2017). During the growing season, this cooling could effectively reduce the degree of





atmospheric sensible heating associated with increased albedo, and would be particularly
important if there is no change in snow masking by vegetation (e.g. greening in tundra without
shrub expansion, or in closed canopy boreal forest). However, the extent to which latent cooling
with enhanced productivity may offset sensible heating associated with albedo decreases is
uncertain for several reasons. First, model experiments simulating shrub expansion, for example,
utilize canopy parameterizations for deciduous boreal tree species, because arctic shrub canopy
physiology has not been thoroughly characterized (e.g. Bonfils *et al.*, 2012). Second, existing
observations indicate an increasing degree of stomatal control on evapotranspiration with
vegetation stature (Eugster *et al.*, 2000; Kasurinen *et al.*, 2014), indicating that LE will not
necessarily continue to increase with climate warming, which is supported by the emergence of
browning trends. Additionally, climatic changes in arctic hydrology are highly uncertain and
likely to vary spatially (Francis *et al.*, 2009), meaning that LE may be limited by hydrology in
some places but not others. Lastly, disturbance processes will also alter surface energy dynamics
through short-term direct impacts on ecosystem structure and long-term impacts on post-
disturbance succession (as described above).

**5 Conclusions**

The effects of climatic change on permafrost across the arctic and boreal biomes will be

strongly affected by terrestrial ecosystem influences on surface energy partitioning.
Relationships between permafrost and climate vary spatially with ecosystems properties and
processes, and these patterns in the relationship between permafrost and climate will change over
time as ecosystems respond to climate. These changes will be driven by surface energy
feedbacks operating on local-, regional-, and global-scales. Complex interactions among many of



these feedbacks create uncertainty surrounding the timing and magnitude of the permafrost
carbon feedback.

Interactions among ecosystem processes are not well understood and represent a key

source of uncertainty in the relationship between permafrost soils and climate. In particular, soil
moisture alters soil thermal conductivity, however the influence of vegetation on soil moisture is
unclear. Future work should seek to elucidate interactions between vegetation and soil moisture.
Similarly, concurrent changes in decomposition rates and the quantity and quality of available
substrate may have strong influences on the insulating effects of the soil organic layer, and
changes in the distribution and productivity of mosses may have similar effects. Improved
understanding of the ecosystem processes influencing soil moisture and thermal properties are
necessary to understand the fate of permafrost.

Holistic understanding of changes in vegetation and ecosystem distributions is another

critically important topic for understanding the fate of permafrost. There has been a strong focus
on graminoid-shrub transitions in tundra ecosystems, yet there are a number of other potential
vegetation transitions, many mediated by disturbance, with equally important implications.
These changes are not spatially isolated, and compounding disturbances will likely become
increasingly common. In addition to vegetation changes, constraining the proportion of
landscapes affected by drying versus waterlogging associated with initial permafrost thaw is
central to predicting both soil organic matter stocks.

Lastly, there is a high degree of uncertainty surrounding the net effects of opposing local

and regional ecosystem feedbacks to permafrost soil temperatures.  Model studies that have
examined the net effects of feedbacks across scales typically focus on one type of vegetation
change (e.g. shrub expansion), and so there is less information regarding interactions among





feedbacks associated with multiple ongoing changes. Continued efforts to understand the fate of
permafrost in response to climate will require integrated analyses of processes affecting
permafrost soil thermal dynamics, changing circumpolar ecosystem distributions, and the net
effects of resulting climate feedbacks operating across a range of spatial and temporal scales.


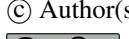



**Acknowledgments**
This project benefited from input from members of the Permafrost Carbon Network
([www.permafrostcarbon.org](http://www.permafrostcarbon.org)). Supporting funding to the Permafrost Carbon Network was
provided by the National Science Foundation Network Grant #955713 and the National Science
Foundation Study of Environmental Arctic Change (SEARCH) Grant #1331083. MML was
supported with funding from the U.S. National Science Foundation grant PLR-1417745. DB was
supported by The Swedish Research Council (2015-00465) and Marie Skłodowska Curie
Actions co-funding (INCA 600398). TAD acknowledges support from the U.S. Army Basic
Research (6.1) Program. BCF was supported by the Academy of Finland (Decision #256991 and
JPI Climate (Decision #291581). IMS received support from UK Natural Environment Research
Council ShrubTundra Grant (NE/M016323/1). Any use of trade, product, or firm names is for
descriptive purposes only and does not imply endorsement by the US Government.



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





Figures

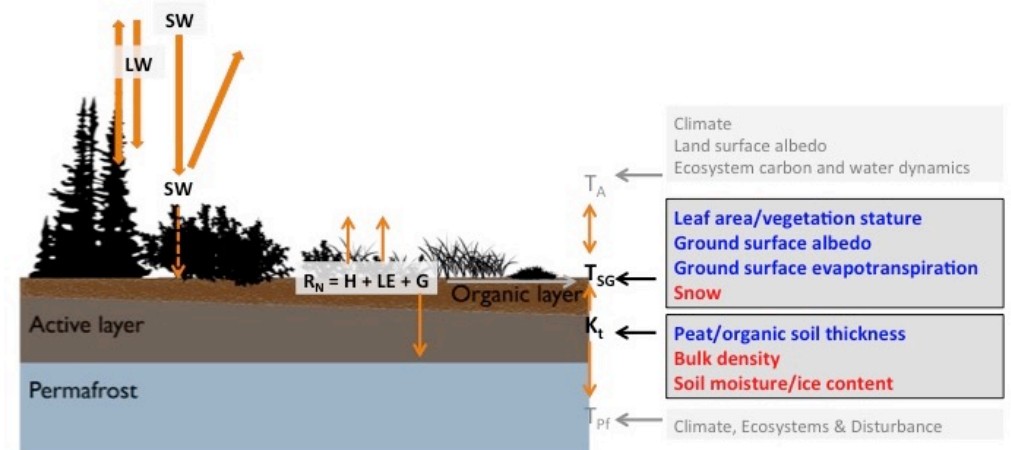

Figure 1. Key ecosystem controls on surface energy partitioning in relation to permafrost soil
thermal dynamics. Net radiation ($R_N$) is balanced by sensible (S) latent (LE) and ground (G) heat
fluxes(energy fluxes are indicated by orange arrows). Ground surface temperature ($T_{SG}$) and soil
thermal conductivity ($K_T$) exert strong controls on G and are strongly influenced by a variety of
ecosystem controls (indicated in dark gray boxes; red and blue text denote soil cooling and
warming effects, respectively). Controls on air ($T_A$) and permafrost ($T_{Pf}$) temperatures are driven
largely by climate, and we assume that ecosystem impacts on these variables are negligible at
short timescales (e.g. season to year) and small spatial scales (e.g. $m^2$ to $km^2$) relative to factors
highlighted in dark boxes.





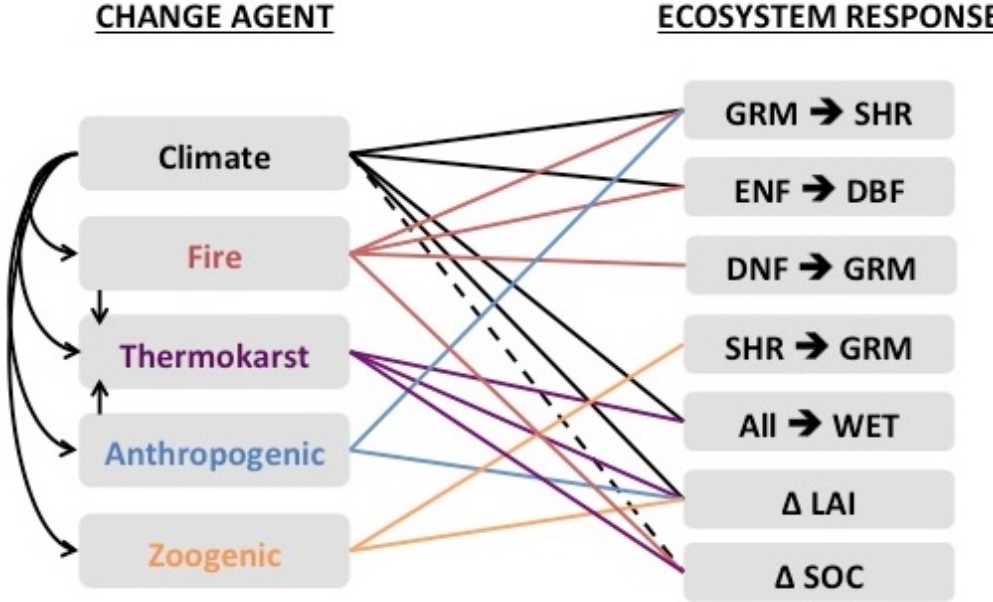


Figure 2. Summary of key drivers of ecosystem change, and the associated ecosystem responses
observed (solid lines) or hypothesized (dashed lines) in permafrost ecosystems. Arrows (**è**)
indicate transition from the current (left) to a new (right) ecosystem type, and the symbol delta
(Δ) indicates a change in the associated ecosystem property. Ecosystem types are defined as
follows: DBF = Deciduous Broadleaf Forest; DNF = Deciduous Needleleaf Forest; ENF =
Evergreen Needleleaf Forest; GRM = Graminoid Dominated Ecosystem; SHR = Shrub
Dominated Ecosystem; WET = Wetland Ecosystem; All = Any Initial Ecosystem type.
Ecosystem properties are: LAI = Leaf Area Index, and SOC = Soil Organic Carbon.



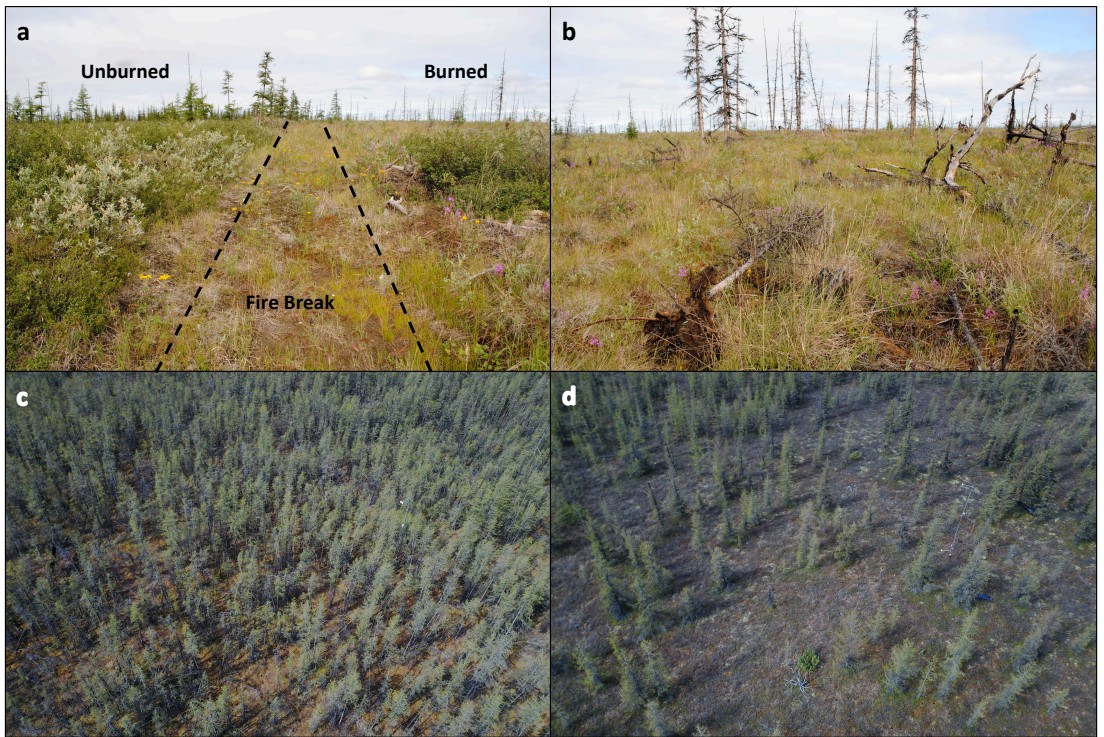

Figure 3. Impacts of fire on ecosystem structure in Siberian larch forests. A firebreak near the
town of Cherskii (a) shows the contrast between burned and unburned areas ~30 years post-fire,
where apparent larch and shrub recruitment failure has resulted a transition to graminoid
dominance (b; detail ofburned area). Nearby in a ~70 year old burn scar high-density (c) and
low-density (d) forests illustrate the impacts of fire severity on canopy cover, and correspond to
large differences in soil thermal regimes and active layers depths (M. Loranty, unpublished data).
Photos M. Loranty.



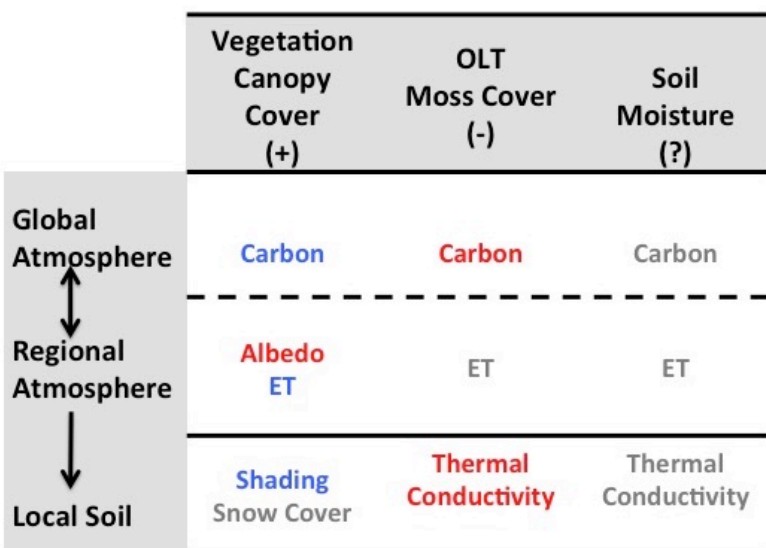

Figure 4. Key ecosystem changes and their associated feedback effects on local soil climate,
regional atmospheric climate, and global climate. The + beneath canopy cover indicates an
assumed increase across the permafrost region, while the – beneath organic thickness and moss
cover indicates an assumed decrease. The change in soil moisture will depend on both changes in
ecosystem-scale hydrologic cycling, as well as changes in regional hydrology driven by climate,
and is assumed to be unknown. Blue text indicates negative feedbacks (cooling effect), red text
indicates positive feedbacks (warming effects), and gray text indicates feedbacks where the
direction is not known.

