# Peer review of "Reviews and Syntheses: Changing ecosystem influences on soil thermal regimes in northern high-latitude permafrost regions"

_Biogeosciences, 2018_

## Referee Comment (RC1) · Anonymous Referee #1 · 31 May 2018

The authors summarize a wide a range of findings on interactions between vegetation, hydrology and soil temperatures in permafrost. A review paper on these complex processes could fill an important gap in the literature. Unfortunately, I am not sure whether the manuscript in its present form achieves this aim. Rather than synthesizing a large spectrum of studies, the manuscript feels disjointed at times. For instance, the impact of hydrological changes is treated separately for winter and summer, thus neglecting important interactions and potential feedbacks. The manuscript also falls short of fulfilling the promise contained in the title, namely the elucidation of the soil thermal dynamics. While the ground heat flux at the soil surface is discussed, many other important aspects of the soil's thermal regime, such as mean permafrost temperatures,

temperature profiles, seasonal amplitudes, ground ice formation, etc., are given very short shrift. I hope that the following comments will be useful to the authors.

1) Thermal dynamics

As stated above, I found the discussion of the soil thermal dynamics incomplete. While the ground heat flux is clearly an important factor, it does not tell the whole story. Also, it is coupled to the subsurface temperature profile, so that is difficult to consider in isolation. These issues are confounded by the fact that the relevant time scales at which the ground heat flux varies are barely discussed. For instance, it is apparently implicitly assumed that the values are averaged over at least a diurnal cycle. Furthermore, the interactions between winter and summer processes are largely left out.

2) Summer and winter-time processes

I felt there was a lack of balance and integration across the annual cycle, and the manuscript thus falls short of its objective to synthesize disparate information. In addition to the problems with the description of the ground heat flux, I had similar reservations about the discussion of the thermal conductivity. I missed a discussion of how the water/ice content modifies the soil thermal conductivity at below-zero temperatures (not explicitly mentioned), and what the impacts on the soil thermal dynamics are. Also, the impact of snow cover on summer-time conditions (soil moisture, deeper soil temperatures, etc.) is not really discussed.

3) Heterogeneity and variability

I believe the co-variability of soil and vegetation properties could be highlighted more clearly, as it has a strong influence on future changes and also on the presently observed patterns of spatial variability. For instance, bryophytes in adjacent wet and dry microtopographical positions often differ greatly in their physical properties. Such interactions can modify observed patterns of e.g. the relation between soil moisture and thaw depths. These issues in interpreting observational (as opposed to experimental

data) are not acknowledged very clearly.

4) Synthesis

I would welcome a greater attempt at synthesizing previous findings, for instance by coming up with testable hypotheses. At present, there are many statements that process X may be important/not important or positive/negative, depending on multiple other factors. By highlighting open questions, or hypothesizing about the most important interactions, the manuscript would be more exciting to read. For instance, the discussion of conductive vs advective heat fluxes would be more informative if the conditions under which large advective contributions are hypothesized to occur (or where they tend to be observed; e.g. in fens in discontinuous permafrost), were mentioned.

Minor issues

1) Energy balance 1

The coupled nature of the surface energy balance and the subsurface dynamics is not portrayed very well. For instance, the following sentence suggests that aboveground processes (rather than above and below-ground processes) determine the surface temperature: 'Once energy has been absorbed at the ground surface and TSG is elevated, soil KT will dictate how much of this energy is transferred downward into the soil'.

2) Energy balance 2

I feel that several important influences of vegetation canopies on the energy balance are neglected (e.g., roughness, longwave radiation from vegetation canopies).

3) line 581

ponding is an important aspect in this context

---

## Referee Comment (RC2) · Anonymous Referee #2 · 11 Jun 2018

Permafrost grounds will undergo pronounced changes in a warmer climate. In the current manuscript the authors focus on how high latitude terrestrial ecosystems influence surface energy fluxes of permafrost soils, and therefore the current soil thermal state and fate of future permafrost degradation. They discuss many aspects of ecosystem/vegetation interactions with the soil thermal regime – interactions which are key to predict future changes in permafrost conditions, but which are not represented (or only represented in a very simplified manner) in current Earth System Models. The authors consider individual processes not in isolation but especially discuss a broad picture of interaction among key processes. Given that current understanding of vegetation-permafrost interactions is incomplete, and that the topic touches on an important as-

pect for model improvement, I consider the paper of broader interest to the readership of Biogeosciences.

Major aspects

1. The multitude of aspects discussed in the manuscript makes it rather difficult for the reader to extract which key processes are likely to govern permafrost-vegetation interactions (under present day conditions and under future climate change). The authors put a lot of effort in discussing a broad spectrum of vegetation-permafrost ground interactions which all influence permafrost soil thermal regimes. Many examples of these interactions reveal the possibility of either a net positive or a net negative feedback, depending on factors such as local topography, climate, soil conditions, etc. A "synthesis" of current knowledge about ecosystem changes and related impacts on permafrost soil conditions would have added value if the discussed aspects of vegetation-permafrost interaction in this manuscript would be summarized such that the reader can judge the broad-scale importance/representativeness of individual processes. In this regard an additional table or figure would be very helpful, which summarizes the discussed aspects in the text and which could list/illustrate

a) the key physical process chains discussed in this manuscript, indicating whether the interactions are likely to result in a net positive of negative feedback (on ground temperatures, or on carbon cycling), or stating that the sign is unclear given current knowledge b) the factors which drive the sign of the feedback (e.g. topography, climate)

To the degree possible, it would also be interesting to illustrate in this table/figure whether feedbacks will rather amplify or dampen under expected Arctic climate change, and (in line with the discussion of fire impacts on page 21) whether changes are reversible or irreversible (on human timescales).

2. A key uncertainty of future high latitude ecosystem changes will come from changes in the hydrologic regime, determined by changes in precipitation, evaporation, and drainage. Projections of these changes are highly uncertain. This aspect should be

discussed in the manuscript as future high latitude vegetation responses will follow rather different trajectories for wetter or drier conditions (compared to today). In this context: Fig. 4 assumes a reduction in future (?) moss cover, and an increase in vegetation canopy cover. What are the assumptions behind made here?

3. One objective of the paper is stated as: " to identify key challenges and research questions that need to be addressed to better constrain how continued climate-mediated ecosystem changes will affect soil thermal dynamics in the permafrost zone."

I might have overseen a discussion of this aspect in the manuscript, but at least in the conclusion section a reference is only made by stating that integrated analyses of processes are needed. A discussion of more concrete aspects would be helpful.

Minor aspects

L61: double occurrence of sentence L 79/80: can you give a reference here? L 126: what is meant by "internal energy transfers"? L 269: Kt depends also on the thermal state (ratio of liquid to frozen water) L688: "available evidence..." can you give a reference here? L 1507: (H) instead of (S) Figure 2: what is meant by "Climate" as change agent – increases in temperature?, what about climate change induced changes in precipitation? Figure 3, L1534: can you give numbers here? Figure 4: OLT is not explained

---

## Author Comment (AC1) · 24 Jul 2018

We thank both reviewers for the insightful and constructive comments, and are happy that they appreciate the value of our review in helping to identify important knowledge gaps regarding relationships between ecosystems and permafrost thermal regimes. The manuscript is greatly improved as a consequence of substantive revisions made in response to these comments. Specifically we have: 1) provided a more thorough and systematic treatment of the ground thermal regime and incorporated this more fully into the overall framework of the review, 2) more fully synthesized the findings of existing studies and identified concrete research questions that need to be addressed, and 3) addressed all of the minor issues. Below we provide our responses to specific reviewer comments. Reviewer comments are shown in blue Times New Roman font, and our responses follow directly in black Cambria font. We indicate where we have revised the manuscript in response to each comment, and provide two versions of the revised manuscript, one with track-changes highlighted, and a second final version with all changes accepted. In our response to specific comments we indicate page and section numbers where revisions are found in the tracked-changes version of the revised manuscript so that the editor and reviewers can easily see them.

**Anonymous Referee #1**

The authors summarize a wide a range of findings on interactions between vegetation, hydrology and soil temperatures in permafrost. A review paper on these complex processes could fill an important gap in the literature. Unfortunately, I am not sure whether the manuscript in its present form achieves this aim. Rather than synthesizing a large spectrum of studies, the manuscript feels disjointed at times. For instance, the impact of hydrological changes is treated separately for winter and summer, thus neglecting important interactions and potential feedbacks. The manuscript also falls short of fulfilling the promise contained in the title, namely the elucidation of the soil thermal dynamics. While the ground heat flux at the soil surface is discussed, many other important aspects of the soil's thermal regime, such as mean permafrost temperatures, temperature profiles, seasonal amplitudes, ground ice formation, etc., are given very short shrift. I hope that the following comments will be useful to the authors.

We are glad the reviewer sees the utility of our paper in filling an important gap in the literature, and appreciate these helpful comments. The manuscript is improved as a result of a more comprehensive inclusion of the ground thermal regime and greater synthesis. Below we respond to specific comments and indicate where we have made changes.

1) Thermal dynamics
As stated above, I found the discussion of the soil thermal dynamics incomplete. While the ground heat flux is clearly an important factor, it does not tell the whole story. Also, it is coupled to the subsurface temperature profile, so that is difficult to consider in isolation. These issues are

confounded by the fact that the relevant time scales at which the ground heat flux varies are barely discussed. For instance, it is apparently implicitly assumed that the values are averaged over at least a diurnal cycle. Furthermore, the interactions between winter and summer processes are largely left out.

In our focus on G as a unifying process and context for considering ecosystem effect s on permafrost it is clear that we failed to comprehensively consider the full thermal regime. Consequently we have modified the beginning of section 2 (p5-7). A new paragraph at the beginning of section 2 explicitly describes the important components of the annual soil thermal regime and how they are quantified. We have also added language to emphasize the importance of above- and below-ground controls on G (p6). Here we should note that in the original manuscript we chose not to emphasize many of these belowground aspects because factors such as the permafrost temperature and the vertical temperature profile are affected by non-ecosystem factors such as long-term climate, geologic and geomorphic history, permafrost genesis, etc., and so are beyond the scope of this review. We also explicitly note our focus on seasonal to annual variability in the soil thermal regime (p 7). Throughout the manuscript processes are now discussed in the context of how changes in G relate back to seasonal and annual aspects of the thermal regime. Interactions between summer and winter are addressed in response to the following context.

2) Summer and winter-time processes
I felt there was a lack of balance and integration across the annual cycle, and the manuscript thus falls short of its objective to synthesize disparate information. In addition to the problems with the description of the ground heat flux, I had similar reservations about the discussion of the thermal conductivity. I missed a discussion of how the water/ice content modifies the soil thermal conductivity at below-zero temperatures (not explicitly mentioned), and what the impacts on the soil thermal dynamics are. Also, the impact of snow cover on summer-time conditions (soil moisture, deeper soil temperatures, etc.) is not really discussed.

Thank you for pointing this out. Section 2 is extensively revised with expanded discussion of wintertime processes and how summer and winter processes are integrated. Sections 2.1 and 2.2 have been combined into a single section focused on canopy processes. It retains the same organization as the previous version, but includes a paragraph at the end discussing integrated effects of canopy processes across the annual cycle, and also identifies clear hypotheses and directions for future research. In sections 2.2 and 2.3 (formerly 2.3 and 2.4) we have also added discussion of seasonal interactions where appropriate. This includes discussion of how water/ice affect thermal conductivity at sub-zero temperatures. In addition we have substantially revised portion of section 2.4 (formerly 2.5) to focus more explicitly on process interactions that impact the soil thermal regime across annual timescales. With this more direct synthesis of results we are able to

offer more specific directions for future research.

3) Heterogeneity and variability
I believe the co-variability of soil and vegetation properties could be highlighted more clearly, as it has a strong influence on future changes and also on the presently observed patterns of spatial variability. For instance, bryophytes in adjacent wet and dry microtopographical positions often differ greatly in their physical properties. Such interactions can modify observed patterns of e.g. the relation between soil moisture and thaw depths. These issues in interpreting observational (as opposed to experimental data) are not acknowledged very clearly.

We have highlighted these points more clearly in our revisions to section 2, and highlighted issues associated with interpreting observational vs. experimental data in our revisions to section 2.5.

4) Synthesis
I would welcome a greater attempt at synthesizing previous findings, for instance by coming up with testable hypotheses. At present, there are many statements that process X may be important/not important or positive/negative, depending on multiple other factors. By highlighting open questions, or hypothesizing about the most important interactions, the manuscript would be more exciting to read. For instance, the discussion of conductive vs advective heat fluxes would be more informative if the conditions under which large advective contributions are hypothesized to occur (or where they tend to be observed; e.g. in fens in discontinuous permafrost), were mentioned.

Thank you for highlighting this. We have revised the manuscript to more clearly identify open research questions and develop hypotheses regarding the directionality and importance of process interactions. These are included at the end of each appropriate paragraphs and sections, and summarized in the conclusions section.

Minor issues
1) Energy balance 1
The coupled nature of the surface energy balance and the subsurface dynamics is not portrayed very well. For instance, the following sentence suggests that above- ground processes (rather than above and below-ground processes) determine the surface temperature: 'Once energy has been absorbed at the ground surface and TSG is elevated, soil KT will dictate how much of this energy is transferred downward into the soil'.

As described above, we have included more thorough discussion of belowground processes as a component of the expanded focus on the soil thermal regime.

2) Energy balance 2

*I feel that several important influences of vegetation canopies on the energy balance are neglected (e.g., roughness, longwave radiation from vegetation canopies).*

We have modified Figure 1 and expanded our discussion of canopy influences on energy partitioning (p9-10) to include these important processes.

*3) line 581 ponding is an important aspect in this context*

Agreed, we have amended the sentence to reflect this.

**Anonymous Referee #2**

*Permafrost grounds will undergo pronounced changes in a warmer climate. In the current manuscript the authors focus on how high latitude terrestrial ecosystems influence surface energy fluxes of permafrost soils, and therefore the current soil thermal state and fate of future permafrost degradation. They discuss many aspects of ecosystem/vegetation interactions with the soil thermal regime – interactions which are key to predict future changes in permafrost conditions, but which are not represented (or only represented in a very simplified manner) in current Earth System Models. The authors consider individual processes not in isolation but especially discuss a broad picture of interaction among key processes. Given that current understanding of vegetation- permafrost interactions is incomplete, and that the topic touches on an important aspect for model improvement, I consider the paper of broader interest to the readership of Biogeosciences.*

Thank you, what you describe is exactly the aim of our review and we are pleased this came across in the manuscript.

*Major aspects*

*1. The multitude of aspects discussed in the manuscript makes it rather difficult for the reader to extract which key processes are likely to govern permafrost-vegetation interactions (under present day conditions and under future climate change). The authors put a lot of effort in discussing a broad spectrum of vegetation-permafrost ground interactions which all influence permafrost soil thermal regimes. Many examples of these interactions reveal the possibility of either a net positive or a net negative feedback, depending on factors such as local topography, climate, soil conditions, etc. A "synthesis" of current knowledge about ecosystem changes and related impacts on permafrost soil conditions would have added value if the discussed aspects of vegetation-permafrost interaction in this manuscript would be summarized such that the reader can judge the broad-scale importance/representativeness of individual processes. In this regard an additional table or figure would be very helpful, which summarizes the discussed aspects in the text and which could list/illustrate a) the key physical process chains discussed in this manuscript, indicating whether the interactions are likely to result in a net positive of negative feedback (on ground temperatures, or on carbon cycling), or stating that the sign is unclear given*

current knowledge b) the factors which drive the sign of the feedback (e.g. topography, climate) To the degree possible, it would also be interesting to illustrate in this table/figure whether feedbacks will rather amplify or dampen under expected Arctic climate change, and (in line with the discussion of fire impacts on page 21) whether changes are reversible or irreversible (on human timescales).

We agree that the manuscript covers so many processes that it is hard to keep track of them all. In order to accomplish this we chose to enhance Figure 4 rather than adding an additional figure or table. Many of the feedback processes were already included in Figure 4, and their linkages to other process were not illustrated elsewhere. So this seemed like a logical place to do this. As we worked through the manuscript we did indeed attempt to create a diagram illustrating all of the process linkages, their impacts on permafrost thermal regimes, key drivers, and associated climate feedbacks. However it quickly became apparent there were simply too many connections to explicitly illustrate them all, for example using block and arrows as we did in Figure 2. Thus we adopted a modified table approach in our revision of Figure 4.

2. A key uncertainty of future high latitude ecosystem changes will come from changes in the hydrologic regime, determined by changes in precipitation, evaporation, and drainage. Projections of these changes are highly uncertain. This aspect should be discussed in the manuscript as future high latitude vegetation responses will follow rather different trajectories for wetter or drier conditions (compared to today). In this context: Fig. 4 assumes a reduction in future (?) moss cover, and an increase in vegetation canopy cover. What are the assumptions behind made here?

Thank you for highlighting this point. We have included discussion of hydrologic uncertainty where appropriate throughout sections 3-5, and in Figure 2. As described above, Figure 4 has been revised to provide more details regarding key changes and feedbacks.

3. One objective of the paper is stated as: " to identify key challenges and research questions that need to be addressed to better constrain how continued climate- mediated ecosystem changes will affect soil thermal dynamics in the permafrost zone." I might have overseen a discussion of this aspect in the manuscript, but at least in the conclusion section a reference is only made by stating that integrated analyses of processes are needed. A discussion of more concrete aspects would be helpful.

Reviewer 1 also raised this concern and we have revised the manuscript to provide more explicit and informative synthesis of the information presented. Within each section we have summarized key process interactions that are poorly understand and where possible hypothesize regarding the likely impact on permafrost thermal regimes. We have also

worked to synthesize key processes across spatial and temporal scales more explicitly, and revised the conclusions to provide a clearer description of the key research challenges and questions.

Minor aspects

L61: double occurrence of sentence

The duplicated sentence has been removed.

L 79/80: can you give a reference here?

Yes, we have amended the sentence and added a reference.

L 126: what is meant by "internal energy transfers"?

We modified this sentence to indicate that internal energy transfers refer to energy fluxes within the soil associated with water phase changes and temperature gradients within the soil.

L 269: Kt depends also on the thermal state (ratio of liquid to frozen water)

The sentence has been amended to reflect this.

L688: "available evidence. . ." can you give a reference here?

This sentence was meant to synthesize information presented in the preceding sentences and has been revised accordingly.

L 1507: (H) instead of (S)

This typo is corrected.

Figure 2: what is meant by "Climate" as change agent – increases in temperature?, what about climate change induced changes in precipitation?

The figure has been amended to indicate climate warming as the driver. In addition we have added a indicating to acknowledge climate induced changes in precipitation, and the associated uncertainty as discussed in our response to your comment above.

Figure 3, L1534: can you give numbers here?

Yes – we have included approximate active layer depths for each site; ~40cm for the high-density and ~90cm for the low-density.

Figure 4: OLT is not explained

This has been addressed through the figure revisions described above.